# ENHANCING CROSS-LINGUAL EMBEDDING ALIGNMENT WITH ADDITIVE KEYWORDS FOR INTERNATIONAL TRADE PRODUCT CLASSIFICATION

## ABSTRACT

Cross-lingual embedding alignment plays an important role in enabling effective multilingual classification tasks. Although multilingual pretrained language models and fine-tuning techniques are increasingly adopted, current approaches inadequately address specialised domains, where domain-specific terminology and mixed-language content present unique challenges that hinder classification accuracy. This work considers the problem of automatically classifying text-based descriptions of international trade transactions with respect to an international standard Harmonized System (HS) code taxonomy. We propose a novel method that incorporates mixed-language keyword embeddings to improve cross-lingual alignment, focusing on bilingual models, and subsequently leverages this alignment for downstream classification tasks, with particular applicability to low-resource domains. Using a supervised learning framework implemented through neural network architectures, the model is trained on pairs of product descriptions and their corresponding extracted keywords. Experimental results on benchmark bilingual datasets demonstrate significant and consistent improvements in classification performance over baseline models, including in low-resource target language scenarios. The findings demonstrate the effectiveness of incorporating additive keywords as a strategy for cross-lingual embedding alignment, thereby enhancing representation quality and improving classification accuracy.

## 1 INTRODUCTION

Language diversity plays a critical role in strategic decisions, particularly in the interpretation of regulations to comply with obligations (Tenzer et al., 2017). In the domain of international trade, it is evident that language differences can hinder bilateral trade flows (Lohmann, 2011). Trade documents in international trade are governed by the Harmonized System (HS) (World Customs Organization, 2018), which presents inherent complexity related to classifying products involving a large number of codes and the use of multiple languages in trade declarations (Novith, 2024; Paramartha et al., 2021). Product descriptions typically follow local language, but traders widely use international languages like English for clarity and industry relevance (Novith, 2024; Paramartha et al., 2021). This treatment is permitted under Indonesian law to allow English as a supplementary language to improve the completeness of information (DGCE, 2009). The presence of this bilingual language in the product declarations complicates the trade facilitation process and highlights the necessity for robust classification systems (Grainger, 2024).

Most existing studies evaluating product classification models have utilised monolingual datasets and mainly in English (Deniss Ruder, 2020; Luppes, 2019; Shubham et al., 2023; Spichakova & Haav, 2020). Monolingual language models limit their applicability in multilingual trade environments, where linguistic diversity is prevalent. On the other hand, non-English datasets have been utilised individually to evaluate model performance in specific regional contexts, such as those in Brazilian Portuguese (de Lima et al., 2022), Korean (Lee et al., 2024), Chilean (de Artiñano et al., 2023) and Chinese (Liao et al., 2024). Notably, the availability of non-English datasets in the public domain is limited due to non-disclosure agreements with data providers.

Several studies utilised a multilingual model by exploiting cross-lingual features for improved performance (Conneau & Lample, 2019; King, 2024). Cross-lingual word embeddings facilitate the transfer of lexical knowledge to align word representations from equivalent text data in embeddings space, which offers benefits for cross-lingual document classification (Ruder et al., 2019). A method for cross-lingual word embeddings uses two monolingual embeddings and bilingual dictionaries to align their embedding spaces with the MUSE model, which presented this approach effectively (Lample et al., 2018). The model has been utilised in a study to classify product titles using German and French languages (Lehmann et al., 2020). Another method uses parallel data consisting of text paired with translations in other languages, demonstrated in the LASER model (Artetxe & Schwenk, 2019). LASER employs an encoder-decoder architecture to generate ready-to-use cross-lingual embeddings that work across the languages it was trained on. The XLM model utilised a Transformer-based architecture for cross-language representation learning through contextual representations rather than word-level translation dictionaries and facilitates fine-tuning for domain-specific applications (Conneau & Lample, 2019). Additionally, a fine-tuning approach to multilingual models, such as multilingual BERT (Devlin et al., 2019), was utilised in dealing with Chilean and Spanish in trade product classification (de Lima et al., 2022).

The characteristics of trade datasets show they are clearly dominated by domain-specific product terminology, such as the brevity of the text length, which often yields insufficient features resulting in poor classification performance (Chuanakrud et al., 2021; Tang et al., 2022). For text classification tasks using short texts, a few words often define the model outcomes (Zhou et al., 2023). Given these aspects, involving keywords of the main product description is potentially supportive in representing product information in a better way for classification tasks. A keyword is a word or phrase that encapsulates the essential information of a text, which potentially serves as a guiding model to emphasise essential features (Blanchard et al., 2022; Chuanakrud et al., 2021).

This study hypothesises that integrating original product descriptions with extracted keywords improves bilingual classification for trade product classification employing HS codes. Utilising a transformer-based language model in this integration and prediction methodology is expected to enhance the representation quality of both text and keywords, which in turn improves the classification performance.

Our main contributions to this paper are summarised as follows.

- We propose a novel configuration of paired training samples consisting of product descriptions and their corresponding cross-lingual keyword features, referred to as mixed keywords. The mixed keywords are generated using a simple yet effective keyword model, which is tailored and subsequently used as a representation of the product category.
- We propose an architecture for a bilingual model consisting of several components of the neural network: a dual encoder, a fusion layer, and deep classification layers, designed for product category prediction (e.g., HS code).
- We propose a bilingual model trained through incremental training to generate a bilingual model that effectively achieves high performance in low-resource settings. The empirical evaluation compared to several established multilingual baselines shows the effectiveness of our proposed method as a potential application for a real-world system.

The paper is structured as follows: first, it outlines the model design, then presents the experimental setup, and subsequently, it presents key results with corresponding analyses, concluding with the main findings.

## 2 METHODOLOGY

Keywords refer to words or phrases that encapsulate the main topic, theme, or essential information within a text and they are valuable in domain-specific contexts where terminology is often highly specialised (Abulaish et al., 2022; Chen, 2024; Nomoto, 2022). Terminology in domain-specific areas, such as trade, legal, or medical domains, is often complex and noisy, making keywords particularly valuable for narrowing the focus to the core problem (Zhang et al., 2021). In the classification of particularly short texts with a limited word count, the results are often determined by just one or two key terms (Zhou et al., 2023). Therefore, keywords can act as discriminative indicators by di-

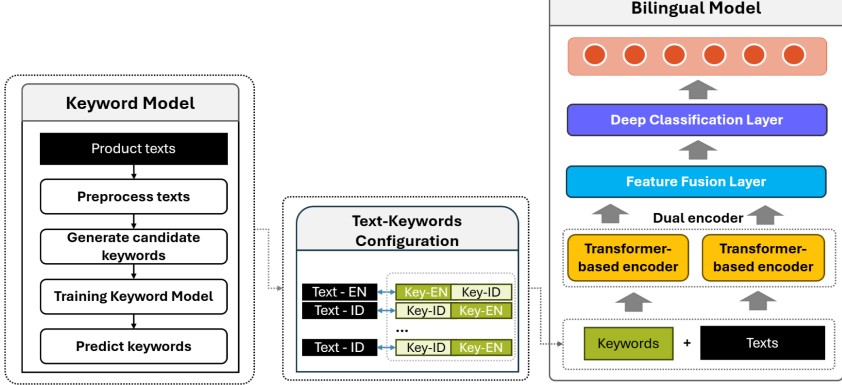

Figure 1: Overview of the methodology for text and mixed keyword integration in bilingual product classification, consisting of: a keyword prediction model, sample configuration, and a bilingual classification model for predicting product categories

recting models toward the most informative features of the text (Blanchard et al., 2022; Chuanakrud et al., 2021).

As such, integrating keywords with the original text can intensify contextual signals, enrich textual representation, and improve model performance, especially when applied to domain-specific datasets. In trade data, product declarations are typically brief, contain mixtures of product attributes, lack grammatical rules, and are intricate to understand the context. The use of keywords in classification tasks is expected to enrich text representation by providing more informative features for effective models' understanding. Keywords integration has the prospect to help in preserving important semantic information from minority classes to improve model generalisation, although this approach does not serve as a primary solution for data imbalance. In bilingual trade datasets, product descriptions are grouped with their respective categories, but they may lack shared semantic alignment due to language-specific differences. Additional keywords become essential to provide higher-level information, acting as semantic anchors that bridge and generalise across category labels.

We extract keywords from the product description and subsequently use them to guide the model and facilitate product representations across different languages, expecting the products belonging to the same category to be similarly represented. Our approach differs from previous studies, such as Onan (Onan et al., 2016), where keywords were used to replace and represent the original texts. This study aligns more closely with the work of Zhou (Zhou et al., 2023), in particular with the strategy of expanding input features by retrieving supplementary knowledge from external open knowledge bases (e.g., DBpedia), which is then fused with the original text to enrich the representation. Our approach is framed within the scope of feature engineering, integrating keywords with primary product descriptions as input for training bilingual classification models. Transformer-based language models are equipped with the capability to recognise nuanced semantic differences in product descriptions, and a neural network-based architecture was employed to learn joint representations from both feature sources, producing semantically aligned product output.

The proposed method integrates product descriptions and extracted keywords into a deep learning architecture for bilingual trade product classification, as illustrated in (Figure 1). It comprises three main components. The early phase addresses keyword extraction, keyword modelling, and text-keyword sample configuration. Next, a bilingual classification model is subsequently constructed through a dual encoder strategy, a feature fusion mechanism, and a deep classification layer.

## 2.1 PROBLEM FORMULATION

We formulate the bilingual-text classification problem as follows. Let $L \in \{l_{\text{EN}}, l_{\text{ID}}\}$ be a set of languages where $l_{\text{EN}}$ represents product description in English and $l_{\text{ID}}$ represents Indonesian, $X = \{x_1^{g_1}, x_2^{g_2}, \ldots, x_z^{g_z}\}$ be a set of product descriptions where $g_i \in L$. Let $C = \{0, 1, \ldots, c\}$

be the categories (i.e., HS codes) of product descriptions. The dataset is denoted by $D = \{(x_1^{g_1}, y_1), (x_2^{g_2}, y_2), \ldots, (x_n^{g_n}, y_n)\}_{i=1}^{n}$ which consists of $n$ samples and $y_i$ an associated label where $y_i \in C$. Let $K_i = \{k_1^{g_1}, k_2^{g_2}\}$ represent the keywords for each product category $y_i$. Given a training dataset $D = \{(X_1, K_1, y_1), (X_2, K_2, y_2), \ldots, (X_N, K_N, y_N)\}_{i=1}^{N}$ the goal of our task is to train a model $F$ that predicts HS code $\hat{y}_i = F((X_i, K_i), \theta)$ where $\theta$ is a set of model parameters, for a given product with a description $X_i$ and the extracted keywords $K_i$. Implementation details and source code: https://github.com/anon-user-temp/mixed-keywords-hscode.

## 2.2 KEYWORD EXTRACTION AND MODELLING

To extract keywords using supervised approaches, it relies on documents that are explicitly labelled with keywords (Siddiqi & Sharan, 2015), while unsupervised methods utilise the statistical and structural characteristics inherent within the text itself. Unsupervised methods are used to extract keywords, such as Term Frequency (Firoozeh et al., 2020), TF-IDF (Hashemzahde & Abdolrazzagh-Nezhad, 2020), YAKE (Campos et al., 2020), and based on Transformer models, such as KeyBERT (Grootendorst et al., 2023). Our study employed a hybrid approach, utilising TF for unsupervised techniques to extract category-specific keywords. These frequently used terms in trade data are assumed to represent each product category best; these keywords were used to annotate the product transactions, rather than embedding them within the main text. Supervised learning was used to train the keywords classifier predicting the keywords from product descriptions needed for configuring training samples.

TF is considered relevant and important for keyword extraction in trade data, where specific terms are more common to represent theme of the products. These high-frequency terms are assumed to be the main product or representation of the declaration. To optimise obtaining the most relevant information in the product, the extraction information or keywords started with text preprocessing, which includes converting text to lowercase and removing duplicates, punctuation, numbers, mixed strings, and extra product details. Word frequency is then computed for each class, and the highest-ranking term is selected as the representative keyword. Following Zaraini's rationale (Zaraini & Yusop, 2025), a keyword-per-language strategy was used to minimise semantic noise and ambiguity, resulting in one representative keyword for each product category in both English and Indonesian. These two keywords are combined to form what we refer to as the Mixed Keywords (MK) strategy. The MK strategy integrates the top keyword from each language, enabling cross-lingual alignment between product descriptions and their associated keywords. These combined keywords are then used to annotate the training data for the keyword prediction model. The detailed procedure is outlined in Algorithm 1 in the Appendix. Having obtained all of the MK labels to represent the product categories, the keyword classifier is designed to predict the product descriptions and their representative keywords. This method used a discriminative modelling approach to assign each text instance a specific keyword label. The classifier employed DistilBERT fine-tuned on an annotated dataset of product descriptions paired with their corresponding keywords. Training followed standard sequence classification procedure (Devlin et al., 2019; Sun et al., 2020), and the model parameters are presented in Appendix Table 6. During inference, predicted MK from the keyword model were combined with their corresponding product descriptions to create paired inputs. These pairs served as inputs for the next phase, in which a bilingual product classification model was trained. The integrated inputs are subsequently processed using deep learning methods to optimise classification performance.

## 2.3 DUAL ENCODER

A separate dual-encoder framework was used to generate representations from sample training consisting of paired product descriptions and MK, which were utilised to fine-tune DistilBERT (Sanh et al., 2020) to generate 768-dimensional embeddings from each encoder. This process is formalised as follows: the product text sequence $X$ is encoded into a latent representation $H_T$ via the encoder function $f_T$. Similarly, the keyword sequence K is encoded by the keyword encoder function $f_K$, resulting in the representation $H_K$. Let $n$ and $m$ denote the maximum sequence lengths of the text and keyword, respectively, and let d represent the embedding dimension. The resulting embeddings are defined as: $H_T = f_T(X) \in \mathbb{R}^{n \times d}$ for product representation and $H_K = f_K(K) \in \mathbb{R}^{m \times d}$ for keyword representation.

## 2.4 FUSION LAYER

For the bilingual classifier, this study employed an intermediate fusion strategy, where the concatenation of product description and MK was performed after the encoding phase, rather than at the raw text level (Boulahia et al., 2021). This decision was motivated to produce richer and more expressive representations to integrate the distinct representations of product text and keywords. To form a unified representation, the contextual embeddings of the $[CLS]$ token from both the text and keyword encoders are concatenated, resulting in a composite vector $h_c$. The operation is formalised as follows: $h_T = H_T [\text{CLS}]$ and $h_K = H_K [\text{CLS}]$ are the [CLS] representations from product descriptions and keywords respectively, and $h_c = \text{Concat} [h_T; h_K] \in \mathbb{R}^{2d}$ is their concatenated joint representation.

The 1536-dimensional concatenated embeddings were processed through a feed-forward layer that projected them to 768 dimensions. This process was followed by layer normalisation to stabilise training by mitigating gradient-related issues and a ReLU activation to introduce non-linearity to learn complex feature interactions, and dropout layers were used to prevent overfitting. The full process is formally described as follows: $f_{\text{Fusion}} : \mathbb{R}^{2d} \to \mathbb{R}^d$ and the output is $f_{\text{Fusion}}(h_c) = \text{ReLU}(\text{LayerNorm}(W_f h_c + b_f))$.

## 2.5 DEEP CLASSIFICATION LAYER

A deep classifier processed the combined text-keyword embedding through three fully connected layers, each activated by ReLU and followed by dropout to reduce overfitting. Training employed cross-entropy loss with label smoothing, replacing hard target labels with soft probability distributions to reduce overconfident predictions and improve robustness to noisy labels (Lukasik et al., 2020), with the parameter set to 0.1. Together, these elements produce a more generalisable and stable model. The full classification pipeline is described as follows: $a^{[0]} = f_{\text{Fusion}}(h_c)$. Subsequently, each layer computes $z^{[l]} = W^{[l]} a^{[l-1]} + b^{[l]}, \quad \text{for} \quad l = \{1, 2, 3\}$, followed by ReLU activation $a^{[l]} = \text{ReLU}(z^{[l]})$. The final output $P(y|X, K; \theta) = \text{Softmax}(z^{[3]})$ provides class probabilities.

# 3 EXPERIMENTS

## 3.1 DATASETS

This experiment utilised international trade datasets in English and Indonesian languages. The English dataset is publicly available and has been widely referenced in existing research (Deniss Ruder, 2020; Luppes, 2019; Shubham et al., 2023; Spichakova & Haav, 2020). These datasets are sourced from publicly available data on the Internet (Enigma, 2018). The Indonesian dataset was obtained from an Indonesian financial institution that has been cited in the literature (Paramartha et al., 2021; Harsani et al., 2020).

The integration of high-resource English datasets with available Indonesian datasets resulted in the matching of 49 distinct product categories, which were identified through the usage of six-digit HS codes. The overview of the dataset statistics, including the total number of samples, is provided in Table 1, with the dataset distribution shown in Figure 3.1. The keyword extraction and modelling method was applied to a random sample of half of each dataset, ensuring all class categories are represented. Table 2 shows examples of product descriptions and their predicted keywords. The performance of the models was evaluated using 10-fold cross-validation (CV-10) using the remaining split of datasets configured in pairs, consisting of product description and the predicted MK, comparing the performance of the proposed model to the baseline models. An ablation study was conducted to evaluate bilingual classification performance of the respective contributions of product descriptions and keywords individually. Additionally, an evaluation scenario was designed to reflect conditions of limited target-language data availability, particularly for non-English datasets. The low-resource setting limited the Indonesian training samples to four levels: 0, 1,000, 5,000, and 10,000. A train-test-holdout split strategy was used for consistent and reliable evaluation.

Table 1: Dataset size in English and Indonesian language.

| Language | Rows |
|----------|------|
| English | 34,937 |
| Indonesian | 19,920 |

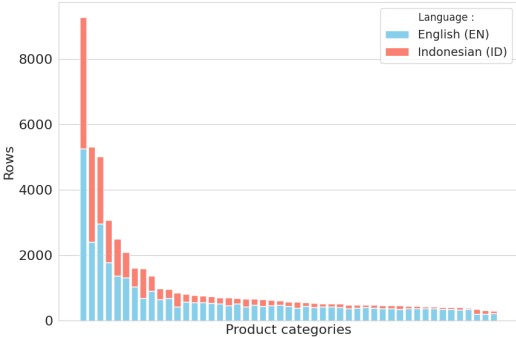

Figure 2: Distribution of bilingual data across product categories

## 3.2 EXPERIMENTAL SETTINGS

### 3.2.1 MODEL BASELINES

To evaluate the effectiveness of the proposed method, we conducted a comparative analysis using several multilingual language models trained with a similar configuration to the proposed method. The baseline models are as follows:

- Multilingual BERT (Devlin et al., 2019): mBERT was pretrained on corpora from 104 languages, enabling it to produce language-agnostic representations that generalise well across multilingual and bilingual settings.

- XLM-R (Conneau et al., 2020): XLM-RoBERTa was built on the RoBERTa architecture, an optimised variant of BERT to generate language-agnostic representations and demonstrates superior results on multiple cross-lingual benchmarks.

- Multilingual DeBERTaV3 (He et al., 2023): mDeBERTaV3 is a multilingual extension of the DeBERTa architecture, outperforming XLM-R by 3.6% across 15 languages.

- Sentence BERT (Reimers & Gurevych, 2019): The model used in this experiment was paraphrase-multilingual-MiniLM-L12-v2, which supports cross-lingual semantic similarity tasks and can be fine-tuned on multilingual datasets.

- IndoBERT (Koto et al., 2020): IndoBERT is a transformer-based model pretrained exclusively on Indonesian datasets and serves as a strong baseline for comparision and classifying using Indonesian text.

### 3.2.2 EXPERIMENTAL SETUP

All experiments and data analysis procedures were executed on P100 GPUs with 16 GB of RAM. The pre-trained models were obtained via the HuggingFace library (Wolf et al., 2020), which preferred the base version and uncased models. The embedding size of all models was set to 768, with the input text or keyword limited to a fixed sequence length of 128 tokens per sample, initiating the truncation and padding process to reach this length. The batch size for training was set to 16, balancing computational efficiency and model performance. The training process employed a learning rate of $2 \times 10^{-5}$ and was conducted over three epochs, which are sufficient for fine-tuning in most tasks. Optimisation used the cross-entropy loss function in conjunction with the AdamW optimiser (Devlin et al., 2019).

Table 2: An example of bilingual product descriptions with extracted mixed keywords.

| Text - EN | Text - ID | Mixed Keywords |
|---|---|---|
| gu06 umbrella alluminium with stone base 60 kgs for outdoor furniture | aksesoris payung taman britam 064 38249 | ['payung', 'umbrella'] |

Table 3: Complete 10-CV test results of all models. The proposed models consistently outperform baseline models across evaluation metrics.

| Models | Accuracy | Precision | Recall | F1-Score |
|---|---|---|---|---|
| **Baseline:** | | | | |
| Multilingual BERT | 0.820 ± 0.006 | 0.819 ± 0.007 | 0.820 ± 0.006 | 0.817 ± 0.007 |
| XLM-R | 0.802 ± 0.007 | 0.801 ± 0.008 | 0.802 ± 0.007 | 0.798 ± 0.007 |
| Multilingual DeBERTa | 0.794 ± 0.008 | 0.793 ± 0.008 | 0.794 ± 0.008 | 0.788 ± 0.008 |
| SBERT | 0.794 ± 0.004 | 0.792 ± 0.005 | 0.794 ± 0.004 | 0.788 ± 0.005 |
| Indo BERT | 0.791 ± 0.005 | 0.790 ± 0.005 | 0.791 ± 0.005 | 0.787 ± 0.006 |
| **Proposed method:** | | | | |
| MK + DistilBERT | 0.970 ± 0.003 | 0.969 ± 0.003 | 0.970 ± 0.003 | 0.968 ± 0.003 |

### 3.2.3 EVALUATION METRICS

The experiment evaluated the model performance using accuracy, weighted precision, recall, and F1-score to handle class imbalance. Additionally, cosine similarity was used to assess semantic alignment between embeddings of cross-lingual product descriptions.

## 4 RESULTS AND DISCUSSION

### 4.1 MODEL PERFORMANCE

The proposed method achieved the highest performance, with an accuracy of 97% and an F1-score of 96.8%, as shown in Table 3. The method also demonstrates performance gain to achieve an average improvement of 17.50% in accuracy and significantly outperforming all baselines; in particular, the best baseline model was achieved by the mBERT model. The F1-score difference was statistically significant, with a large effect size of Cohen's d = 24.90, as shown in Table 5 in the Appendix.

The empirical results are consistent with previous research (Onan et al., 2016; Zhou et al., 2023), which highlights the importance of using keyword features in classification tasks. Furthermore, our approach advances previous studies employed in bilingual models by configuring cross-lingual keywords paired with the main text to support model understanding, which in turn improves the bilingual classification task.

### 4.2 EMBEDDINGS ALIGNMENT

PCA was used to visualise in low-dimensional embeddings to provide insight into the cross-lingual alignment. We only compared the embeddings generated by the XLM-R model and the proposed method. Figure 3.a shows a scattered fine-tuned XLM-R, where both points representing across languages were unclustered. On the other hand, the proposed method generated data points that present a structure overlap between languages, implying improved cross-lingual alignment, illustrated in Figure 3.

In order to further examine the result, a quantitative evaluation was performed the cross-lingual alignment by calculating the average similarity between embeddings in different languages across all product categories. Figure 3.b visualises these scores as a heatmap, where diagonal values indicate intra-class similarity and darker blue shades indicate higher similarity. The metric is calculated as: $\sigma_c^{\text{mean}} = \frac{1}{n_c m_c} \sum_{i=1}^{n_c} \sum_{j=1}^{m_c} \text{sim}_{\cos ij}^c$. Our proposed method achieved better cross-lingual alignment than XLM-R, with an intra-class similarity of 0.92. This result demonstrates that integrating

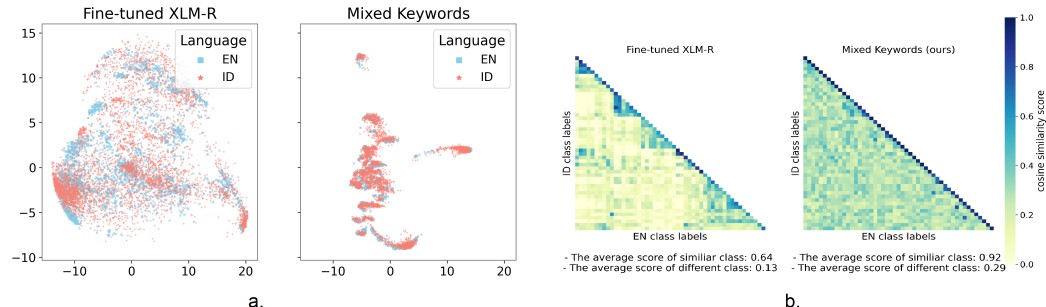

Figure 3: a. PCA visualisation presents cross-lingual alignment in of English (EN) and Indonesian (ID) generated by fine-tuned XLM-R and our proposed method. b. Heatmap shows similarity scores between inter-class and intra-class labels (the diagonal) across ID (y-axis) and EN (x-axis) generated by the fine-tuned XLM-R and the proposed method.

```
Query: rattan table

Top 2 most similar sentences in EN corpus:

940381 : rattan bar set model no8361 1 set with 3 pcs (Score: 0.8470)
940381 : day bednyaman synthetic rattan (Score: 0.8469)

Top 2 most similar sentences in ID corpus:

940381 : tempat lampu rottan kombinasi besi &kain (Score: 0.8437)
940381 : top meja rotan rangka besi (Score: 0.8431)
```

Figure 4: Retrieved product descriptions using bilingual queries.

keywords improves semantic alignment, encouraging product descriptions within the same class to map closely in the embedding space across languages.

We also performed a case study to evaluate the bilingual model by submitting a query of "rattan table". The result shows that the model retrieved the two most relevant products in both languages, correctly mapped to HS code 940381 for furniture made of rattan or bamboo (World Customs Organization, 2018), as seen in Figure 4. The bilingual product search experiment has resulted in cross-lingual product results and captured material information embedded in the queries.

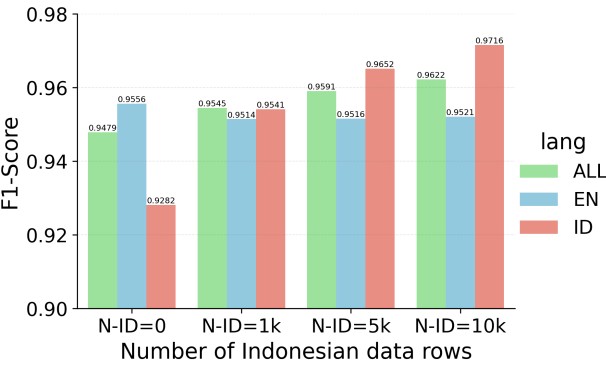

Figure 5: The model's performance on bilingual classification in an incremental training sample setting.

## 4.3 MODEL PERFORMANCE IN LOW-RESOURCE SETTING

Given the scarcity of trade data in non-English languages, and in this study, the Indonesian language was considered a low-resource language. In this experiment, we assumed only to train the model on English training samples and the predicted keyword, or MK. The result shows that the model achieved a notable F1-score of 94.79%. Additional Indonesian training samples used for model training improved the model's performance, achieving a significant F1-score of 96.22%, as shown in Figure 5. This finding demonstrates that combining high-resource languages, such as English, with cross-lingual keywords is sufficient to achieve strong model performance, particularly when the target language is limited or non-existent.

Our implementation of the DistilBERT model (Sanh et al., 2020) in dealing with low-resource language expands on previous research using BERT-based models in low-resource settings (Cruz & Cheng, 2020; Li et al., 2020). DistilBERT is a language model trained on monolingual corpora in English; yet it demonstrates effectiveness on domain-specific data, such as bilingual trade datasets. The model also offers lower computation due to its language model size compared to other multilingual models.

## 4.4 ABLATION STUDY

In this ablation study, comparisons were performed between the proposed model and two variant models when trained only with the product description, specifically Product Only (PO), and the model trained only with predicted keywords, specifically Keyword Only (KO). The results show that the proposed method surpasses both PO and KO by 15.88%–16.04% and KO by 6.83%–10.65%, respectively, evaluated on 10-CV, as shown in Table 4. These results reinforce the concept that relying solely on texts or keywords is insufficient to achieve a high-performance model, and underscore the value of combining keywords with the main product text.

Table 4: The ablation results compare Product Only (PO) and Keyword Only (KO) models, with the lower panel showing the performance gains of the Mixed Keyword (MK) method over both baselines.

|  | Accuracy | F1-Score |
|---|---|---|
| *Models:* | | |
| PO | 0.816 ± 0.005 | 0.813 ± 0.005 |
| KO | 0.904 ± 0.000 | 0.865 ± 0.001 |
| MK | 0.970 ± 0.003 | 0.968 ± 0.003 |
| *Performance Gains:* | | |
| MK vs. PO | +15.88% | +16.04%* |
| MK vs. KO | +6.83% | +10.65%* |

## 5 CONCLUSION

In conclusion, our bilingual models have addressed the issue of bilingual language presented in trade product declarations by developing a bilingual classification model that incorporates cross-lingual keyword features used in the bilingual model training. Mixed keyword strategies facilitated the capture of product terms, addressed language variations, and aligned bilingual trade data. The model successfully enhances the embedding alignment of bilingual languages, demonstrates strong effectiveness in low-resource target language settings, and surpasses the other baseline models, indicating potential applicability in real-world applications. This study was conducted using English and Indonesian data, and the limited availability of the other languages trade datasets presents a generalisation challenge. However, this also suggests a promising avenue for future research to expand the approach to broader multilingual settings.

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

# A APPENDIX

## A.1 MIXED KEYWORDS EXTRACTION ALGORITHM

---
**Algorithm 1** Mixed Keywords Extraction Algorithm

---
**Input**: $Docs_{EN}$, $Docs_{ID}$, product_categories
**Output**: $MK$ (Mixed keywords for each class)

1: **for** each $c$ in product_categories **do**
2:     *// Aggregate EN keyword candidate*
3:     $V_{EN} \leftarrow \{\}$
4:     **for** $d$ in $Docs_{EN}[c]$ **do**
5:       $v \leftarrow$ preprocess($d$)
6:       $V_{EN} \leftarrow$ merge($V_{EN}, v$)
7:     **end for**
8:     $K_{EN} \leftarrow$ sort($V_{EN}$)
9:     *// Aggregate ID keyword candidate*
10:     $V_{ID} \leftarrow \{\}$
11:     **for** $d$ in $Docs_{ID}[c]$ **do**
12:       $v \leftarrow$ preprocess($d$)
13:       $V_{ID} \leftarrow$ merge($V_{ID}, v$)
14:     **end for**
15:     $K_{ID} \leftarrow$ sort($V_{ID}$)
16:     $k_{EN} \leftarrow$ top($K_{EN}$, 1) *// EN keyword*
17:     $k_{ID} \leftarrow$ top($K_{ID}$, 1) *// ID keyword*
18:     $MK[c] \leftarrow [k_{EN}, k_{ID}]$ *// Mixed keywords*
19: **end for**
20: **return** $MK$

---

## A.2 STATISTICAL MEASUREMENT OF THE MODELS

Table 5: Statistical comparison between the proposed MK + DistilBERT model and various multilingual baseline models.

| Baseline Model | p-value | Effect Size (Cohen's d) |
|---|---|---|
| Multilingual BERT | $< 0.001$ | 24.90 |
| XLM-R | $< 0.001$ | 25.40 |
| Multilingual DeBERTa | $< 0.001$ | 24.18 |
| SBERT | $< 0.001$ | 42.34 |
| Indo BERT | $< 0.001$ | 32.65 |

## A.3 SUMMARY OF HYPERPARAMETER CONFIGURATION

Table 6: Model hyperparameters and training configuration.

| Hyperparameter | Value |
|---|---|
| Embedding dimension | 768 |
| Max length tokens | 128 |
| Epoch | 3 |
| Batch size | 16 |
| Learning rate | $2 \times 10^{-5}$ |
| Dropout | 0.2 |
| Optimizer | AdamW |
| Activation Function | ReLU |

