# OpenReview forum: "Enhancing Cross-Lingual Embedding Alignment with Additive Keywords for International Trade Product Classification"
_ICLR.cc/2026/Conference — Submitted to ICLR 2026_

### Official Review · Reviewer_epe2 · 2025-10-25

**Soundness:** 3
**Presentation:** 3
**Contribution:** 2
**Rating:** 4
**Confidence:** 4

**Summary:**

This paper describes an architecture and experiment in NLP for a low resource language -- Indonesian.  The main task is to classify products using Indonesian text.  The paper describes an architecture that builds on DistilBERT, and fuses with an Indonesian models focusing on extracted keywords.  This method works very well in the one experiment, and I think could generalize to other low-resource languages.  Notably it outperforms mBERT, an Indonesian specific model, and other obvious baselines.

**Strengths:**

Extended amazing success of English LLMs to much lower resource languages is a key challenge the ML community is facing.  The idea of focusing on the better represented features (high term frequency words) to fuse the English and Indonesian models is something that might generalize to other languages.

**Weaknesses:**

On the downside, this paper is really about a single experiment on one data set.  There is not evidence that it will be a broadly general approach.  It is not just that it is only on a single language, but it is only on the product classification task.  Half the gains come from careful treatment of the products.

I have mixed thoughts on this.  I think the problem is important, the result on the one experiment is very strong, and it may generalize -- but because it is more limited in experiments than a traditional ICLR paper, I have low confidence to advocate for the paper.

**Questions:**

Can you provide evidence that this approach applies more generally to problems with low resource language?

---

### Official Review · Reviewer_ysaW · 2025-10-29

**Soundness:** 3
**Presentation:** 3
**Contribution:** 2
**Rating:** 4
**Confidence:** 3

**Summary:**

This paper is about cross-lingual embedding alignment, in particular for english and indonesian languages in the setting of product classification. The authors propose a model that combines inputs for product descriptions and keywords to classify products in a setting of international trade.

The model that the authors build is bilingual, able to make classification in english or indonesian.

The contributions are:
- A training process for a model taking two inputs, product descriptions and cross-lingual keywords to represent product categories.
- A bilingual model (english and indonesian) and architecture consisting of dual encoders, fusion layers and classification layers. This model achieves good performance in low resource settings.

**Strengths:**

- The paper is well written and easy to follow.
- The problem seems to be difficult, to classify a product according to a trade standard based on description/keywords in english and local language (indonesian in this case), so this seems to be a real-world problem that can produce insights in ML and linguistics.
- The evaluation seems to be correct (except for results in Figure 5), there is a good selection of baselines, and results on one dataset of english and indonesian products. Overall the conclusion that the model outperforms the baselines seems to be correct.
- There are ablation results with the different input languages and the two inputs (product and keyword or both), which shows the advantage of using both inputs.
- The model seems to perform well in a low resource setting.

**Weaknesses:**

- The bilingual model architecture does not seem novel to me, it looks like a common architecture for cross-lingual modelling. I do not see this as a strong negative, just that the authors claim this architecture as a novelty.
- The results presented in Figure 5 are a bit puzzling, as this experiment varies the number of data points in indonesian language, including zero data points, which I would expect to obtain close to random (50%?) performance since the model does not know indonesian language, but it obtains around 92% F1, which the authors do not explain, how is it possible if the model has not been trained in indonesian language?
- The dataset used for training and evaluation is not clear, there are multiple references pointing to use of a dataset, but not a clear dataset name, the text  in lines 252-256 seems to hide which dataset is being used, please make this information very clear for reproducibility and trustworthiness.
- I believe the results are not really surprising, the authors do mention that results are consistent with the literature (lines 359-363) about importance of keyword features, I believe the authors should highlight the novelty of this paper relative to the state of the art.

Some minor improvements:
- Table 4, there is no need to use abbreviations since there is plenty of space in the table for PO, KO, MK, etc.
- Figure 4, the text looks blurry since it is presented in a raster image, always present text as text, not as images. Additionally I would recommend to include multiple qualitative examples for completeness, similarly for Table 2, these qualitative examples are interesting for the reader.
- Figure 3 would be best presented as vector figures (PDF) from the source.

**Questions:**

- Can you explain the results in Figure 5, specifically the first set of results with zero indonesian data points? How can the model score much higher than random chance if has not been trained in the indonesian language?
- Can you clarify the novelty with respect to the state of the art?

---

### Official Review · Reviewer_khEL · 2025-11-01

**Soundness:** 3
**Presentation:** 3
**Contribution:** 2
**Rating:** 6
**Confidence:** 2

**Summary:**

This work focus on the international trade product classification. The aim of this problem it to predict the HS code based on the mixed-language texts in trade documents. The authors propose a framework to tackle this problem. It consists of keywords extraction and configuration, in which the keywords in English and Indonesian are paired to mixed keywords (MK) as additive information for product classification. Experiments and ablation demonstrate the effectiveness of the proposed framework.

**Strengths:**

1. The task of international trate production classfication is a real application with technical challenges.
2. The paper is well-written and easy to follow. The challenges of international trade product classfication is clearly introduced.
3. The proposed framework is reasonable and can intuitively address the chanlleges that the documents are in mixed languages.
4. The experiments are comprehensive and ablation is valid to demonstrate the effectiveness of the framework.

**Weaknesses:**

1. The scope of this research is a bit limited, and experiments only include English and Indonesian languages.
2. The proposed framework is not novel. The keyword extraction and configuration steps are reasonable but straighforward. The classification module is also a natural design. The overall framework is more like a standard solution.

Overall, the limited scope and incremental design in this work is obvious. My concern is that how much insights this work could provide to a broader community. The contribution of this work can be futher improved with either datasets contribution, such as new datasets on international trade, or in-depth study on international trade scenarios in more languages or more fundamental challenges. But those improvements can be out of the scope of this work, and I acknowledge the contribution in the current manuscript as mentioned above in strength section and would recommend a positive rating.

Minor comments:
1. Illustration and examples can improve the readability as international trade is a vertical application for cross-lingual embedding, as far as I know.
2. Is the proposed approach related to word alignment task [1] in machine translation?

[1] https://en.wikipedia.org/wiki/Bitext_word_alignment

**Questions:**

See weakness section.

---

### Official Review · Reviewer_4G7R · 2025-11-02

**Soundness:** 2
**Presentation:** 3
**Contribution:** 2
**Rating:** 2
**Confidence:** 3

**Summary:**

This paper studies on using mixed-language keywords to enhance cross-lingual embedding alignment for a challenging, real-world task: bilingual HS code classification. The proposed method demonstrates consistent improvements over baselines.

**Strengths:**

-  The proposed method is intuitive and demonstrates performance improvements over several multilingual baselines on bilingual trade classification tasks.
- The method also shows effectiveness in low-resource scenarios, addressing a key limitation in current multilingual classification systems.

**Weaknesses:**

- Limited Justification of Cross-Lingual Alignment: Given the strong inherent capabilities of modern multilingual LLMs to process and align multilingual text, focusing on explicit cross-lingual embedding alignment may have constrained practical value. State-of-the-art LLMs often natively achieve effective cross-lingual understanding and representation.
- The baselines used in the paper are primarily BERT and its variants, which are increasingly outdated. The paper lacks comparative analysis with the latest multilingual and reasoning-oriented LLMs (e.g., Qwen, LLaMA), making it difficult to convincingly demonstrate the proposed method's advantage relative to the current state-of-the-art.
- Experiments are confined to the English-Indonesian language pair and the specific domain of international trade product classification. The method’s effectiveness and transferability to other language pairs, more diverse multilingual settings (such as code-switching or morphologically rich languages), and applications remain uninvestigated.
- The keyword extraction relies heavily on frequency-based methods (e.g., TF), which may miss nuanced or context-specific terms crucial in complex real-world trade data.

**Questions:**

N/A

---

### Meta-Review · Area_Chair_ysbX · 2026-01-05

**Summary:**

The paper addresses the problem of bilingual International Trade Product Classification (specifically HS codes) using English and Indonesian text. The authors propose a method involving "Mixed Keywords" (MK), where keywords are extracted using Term Frequency (TF) from both languages and paired with product descriptions. These are processed through a dual-encoder architecture (based on DistilBERT) with a fusion layer to align cross-lingual embeddings. The study claims that this feature engineering approach significantly outperforms standard multilingual baselines (like mBERT and XLM-R) in classifying trade documents, particularly in low-resource settings.

**Rebuttal Conclusion**:
The authors did not participate in the rebuttal phase and failed to address any of the reviewers' questions or concerns. Consequently, all issues raised regarding novelty, scope, and technical validity remain outstanding. Given the generally low scores and the lack of defense against valid criticisms, the general agreement is to reject the paper.

**Reviewer Concerns:**

**Concerns Addressed by the Rebuttal**
•**None**: As no rebuttal was submitted, no concerns were addressed.

**Outstanding Concerns**:
•**Limited Methodological Novelty**: A primary concern shared by Reviewers khEL and ysaW is that the proposed framework is incremental. The use of standard keyword extraction (TF) combined with a dual-encoder architecture is viewed as a standard engineering solution rather than a novel contribution suitable for ICLR. Reviewer 4G7R also noted that modern LLMs perform cross-lingual alignment natively, limiting the conceptual value of explicit embedding alignment via keyword engineering.
•**Narrow Scope and Generalizability**: Reviewers 4G7R, khEL, and epe2 heavily criticized the limited experimental scope. The paper relies entirely on a single dataset involving one language pair (English-Indonesian) in a specific domain. There is no evidence provided that the method generalizes to other language pairs, broader multilingual settings, or different tasks.
•**Outdated Baselines**: Reviewer 4G7R pointed out that the baselines (BERT, XLM-R, DistilBERT) are becoming outdated. The paper lacks comparison with modern Large Language Models (e.g., LLaMA, Qwen) which represent the current state-of-the-art in multilingual understanding, making the performance claims less convincing.
•**Unexplained Results / Potential Flaws**: Reviewer ysaW raised a critical technical question regarding Figure 5, where the model achieves ~92% F1-score with zero Indonesian training data. This result appears weird or indicative of data leakage/methodological error, and without a rebuttal explaining how the model classifies a language it hasn't seen with such high accuracy, the results are considered untrustworthy.

**Reviewer Scores:**

The general feeling ismostly negative to borderline: 2, 6, 4, 4.
Reviewer 4G7R (Score: 2) recommended rejection due to outdated baselines and limited justification. Reviewers ysaW and epe2 (Score: 4) found the paper marginally below the threshold due to the single-experiment nature and lack of novelty. Only Reviewer khEL (Score: 6) was marginally positive but explicitly stated they would not mind if the paper is rejected given the incremental design.

---

### Decision · Program_Chairs · 2026-01-26

Reject